# Novel High-Resolution Lateral Dual-Axis Quad-Beam Optical MEMS Accelerometer Using Waveguide Bragg Gratings

**Balasubramanian Malayappan** [1] and **Narayan Krishnaswamy** [2] and **Prasant Kumar Pattnaik** [1,*]

[1] Department of Electrical and Electronics Engineering, BITS-Pilani, Hyderabad Campus, Hyderabad 500078, India; balasubramanian@hyderabad.bits-pilani.ac.in
[2] Department of Electronics and Communication Engineering, Sai Vidya Institute of Technology, Bengaluru, Karnataka 560064, India; narayank101@gmail.com
* Correspondence: pkpattnaik@hyderabad.bits-pilani.ac.in; Tel.: +91-40-66303612

**Abstract:** A novel lateral dual-axis a-Si/SiO$_2$ waveguide Bragg grating based quad-beam accelerometer with high-resolution and large linear range has been presented in this paper. The sensor consists of silicon bulk micromachined proof mass suspended by silica beams. Three ridge gratings are positioned on the suspending beam and proof mass to maximize sensitivity and reduce noise. Impact of external acceleration in the sensing direction on the Bragg wavelength of gratings and MEMS structure has been modelled including the effects of strain, stress and temperature variation. Acceleration induces stress in the beam thus modifying the grating period and introducing chirp. The differential wavelength shift with respect to reference grating on the proof mass is the measure of acceleration. To compensate for the effect of the weight of the proof mass and increase the sensitivity of the sensor, electrostatic force of repulsion is applied to the proof mass. For the chosen parameters, the designed sensor has a linear response over a large range and a sensitivity of 30 pm/g. The temperature of surroundings, which acts as noise in sensor performance is compensated by taking differential wavelength shift with respect to reference grating. By design and choice of material, low cross-axis sensitivity is achieved. The proposed design enables a high-resolution well below 1 $\mu$g/$\sqrt{\text{Hz}}$ and is suitable for inertial navigation and seismometry applications.

**Keywords:** accelerometers; high-resolution; quad-beam MEMS; differential measurement; coupled-mode theory; nano-grating

## 1. Introduction

Microelectromechanical systems (MEMS) technology finds various applications because of low cost, small size, ease of fabrication and reliable performance. Sensing is arguably the most prominent application of MEMS with branches that have developed into full-fledged fields like RF-MEMS, BioMEMS, microfluidics and optical MEMS. MEMS sensors based on capacitive, piezoelectric and piezoresistive effects are limited in their performance due to electromagnetic interference (EMI). Capacitive and piezoelectric sensing requires that the electronic sensing must be placed nearby the sensor itself. Integrated sensors using optical readout methods, however, are unaffected by EMI and also do not add to the electromagnetic radiation acting as noise for the other sensors and devices in the vicinity, thus improving device performance [1–5]. Optical readout techniques offer very high sensitivity and resolution.

Among optical components, Bragg gratings are widely used for various sensing applications because their response characteristics can be made sensitive to different external conditions like

temperature, material, stress and pressure. Bragg gratings act as guided-wave optical reflection filters. The filter characteristics from centre frequency/wavelength, full-width at half maximum (FWHM), main lobe power and side-lobe powers are dependent on the properties of the gratings and its surroundings. Coupled-mode theory is used to analyze the Bragg grating properties. By design, Bragg gratings are not point sensors as their response depends on the conditions throughout its length. This makes Bragg gratings and particularly fibre Bragg gratings (FBG) an economically attractive solution for various sensing applications [6–8]. Optical MEMS sensors based on the interaction of MEMS structure with free space light or light in the optical fibre are limited in scope as they cannot be subject directly to vibration or acceleration. However, the integration of optical devices with MEMS enhances the performance of the sensor. Photonic-Integrated-Circuits (PIC), which hold promise of vastly improved performance with much reduced cost and power consumption by miniaturization of optics, are rapidly maturing for silicon platforms with standardization of various photonic library components [9]. Optical MEMS based PICs, which rely on a change in optical properties of a waveguide by electromechanical actuation, have been found to have excellent potential for scalability due to low optical loss and device footprint in addition to low power consumption [10]. Thus, optical MEMS sensors based on integrated light guidance are an attractive avenue for further research and development.

An accelerometer is a sensor used to measure acceleration forces and are used for applications ranging from seismic prospecting, vibration measurement, constant/static force measurement to inertial navigation. Accelerometers are needed in several areas, where the change in speed or vibration in an object needs to be monitored or measured. Quad-beam-mass accelerometer is a popular MEMS accelerometer design commonly used with piezoresistive sensing elements [11]. In the quad-beam accelerometer design, due to acceleration experienced by the device frame, stress is introduced into the suspending beams. By integrating sensing components to the beams, the acceleration is measured. Table 1 presents a summary of various optical sensing techniques for accelerometers. Encoding of acceleration in intensity of the output optical signal is a simple and efficient mechanism for optical sensing suitable for harsh environmental conditions. In this technique, movement of proof-mass either modifies the coupling between light input and output paths [12–14] or blocks the optical path [15,16]. It is a process amenable to integration into PIC. However, these designs have poor fundamental resolution limits. Integrated optical MEMS accelerometers based on microring resonators and racetrack resonators with a sensitivity of 0.015 μm/g have been proposed for seismic prospecting [17,18]. The dynamic range of ring resonator based sensors is limited by the small displacement range over which optical coupling takes place. A high-resolution integrated resonant accelerometer based on an optical microdisk resonator was demonstrated in [19] where the photoelastic effect caused a shift in resonance frequency. To increase sensitivity, the microdisk was integrated with resonant tether of resonant accelerometer instead of the suspending arm. By using a narrow linewidth laser locked to an operating wavelength close to the resonance peak, a linear range of operation with a DC sensitivity of 6 μg/$\sqrt{\text{Hz}}$ for device natural frequency of 16.3 kHz has been obtained. In [20,21], fibre is drawn and wound into microrings, which are positioned on sensitive regions of MEMS structures. This eliminates the need for light input and output coupling. For ring resonator based sensors, the acceleration is encoded in the resonance wavelength of the ring.

**Table 1.** Commonly used optical sensing techniques for accelerometers.

| Sensing Technique | Sensitivity | Range | Bandwidth | Remarks |
|---|---|---|---|---|
| Intensity Modulation [12–16] | Low | Low | Medium | Amenable to integration; free-space light propagation; source noise is crucial |
| Ring resonators [17–21] | High | Low | High | Small footprint; non-linear sensor output; guided-wave propagation; wavelength shift due to acceleration |

**Table 1.** *Cont.*

| Sensing Technique | Sensitivity | Range | Bandwidth | Remarks |
|---|---|---|---|---|
| Photonic Crystals [22,23] | High | Low | High | Coupling light into and out of sensor is challenging; guided-wave propagation; photonic bandgap is modified |
| Diffraction Gratings [24] | High | Low | Low to High | Free-space light propagation; complex packaging; high sensitivity requires complex detection mechanism |
| Interferometers [3,25–29] | Low to High | High | Low to High | Free-space light propagation; precise positioning and alignment needed |
| Bragg gratings [30–36] | Medium | High | Low | Guided-wave propagation; wavelength-encoded acceleration; advanced interrogation schemes allow sub-pico strain detection |

In [22], the authors demonstrated a high-resolution integrated accelerometer using photonic-crystal nanocavity. A resolution in the order of $ng/\sqrt{Hz}$ has been demonstrated based on the coupling of the optical field from delicately tapered fibre held in the evanescent region of the photonic-crystal zipper cavity. In [24], a nano-*g* accelerometer based on sub-wavelength diffraction gratings has been demonstrated. Although by optimizing the detection circuit, resolution close to the limits of optical sensing that were shown for this sensor requires complex packaging. Fabry-Pérot interferometric cavities formed with MEMS structures are popular for accelerometers as the accelerometer proof-mass itself can act as a reflecting end-face to form the cavity. The sensitivity of this class of sensors has been enhanced by increasing mechanical sensitivity [3,27–29] and moving from intensity based detection to coherence demodulation [25] to phase-generated-carrier (PGC) modulation [26]. As the FP cavity length is mechanically modified due to acceleration, this method is based on free-space or unguided propagation of light which poses challenges in terms of quality of alignment and packaging and incurs high optical losses.

Fibre Bragg grating (FBG) based accelerometers have been extensively researched and have been shown useful for high sensitivity measurement over a large range [30,32,33]. High resolution FBG based accelerometers have been demonstrated when used in combination with other resonant structures like ring resonators [37], a Fabry-Pérot (FP) cavity [38], Π-shifted fibre Bragg gratings (PSFBG) [35], ultrastable lasers [34,39] and actively locked lasers using radio-frequency modulation [35,40]. Though silicon-on-insulator (SOI) based planar waveguide Bragg gratings have been applied to other sensing applications like pressure sensors [41,42], the integration of silicon based Bragg grating into PIC has not matured after the initial excitement based on silicon-nitride waveguides [31]. Polymers are an attractive option for planar optical MEMS sensors as waveguides and Bragg gratings can be written directly onto a substrate. Polymer planar Bragg gratings based on materials like polymethylmethacrylate (PMMA) and TOPAS® have been used to demonstrate strain sensors [36,43]. Properties of polymer Bragg gratings depend on humidity apart from temperature and strain and humidity immunity is important for robust performance [44]. The major challenge with using Bragg gratings as strain sensors/accelerometers is that for high resolution (requires sharp Bragg peak), long grating length is required. Thus, the integration of planar waveguide Bragg gratings into MEMS requires relatively large structures. Waveguide Bragg grating based accelerometers are suited for high sensitivity quasi-static acceleration measurements like in seismometry. Since PIC components are generally based on a silicon/SOI platform, we present a planar Bragg grating high-resolution dual-axis accelerometer on silicon substrate.

In this paper, we present the design and analysis of high-resolution integrated waveguide Bragg grating based optical MEMS quad-beam lateral two-axis accelerometer with a large linear range.

The silicon wafer based accelerometer has silicondioxide beams and amorphous-silicon(a-Si) optical waveguide and gratings on these beams for the sensing elements. In our earlier works [45,46], we reported a simulation model of single axis accelerometer based on waveguide Bragg gratings. In this work, the design of a novel configuration of lateral quad-beam optical MEMS accelerometer for multi-axis sensitivity with suppressed cross-axis sensitivity and immunity to environmental effects is presented. Further, detailed analysis of the mechanical and optical response of the proposed optical MEMS device has been done. In Section 4, plausible fabrication steps for realizing the proposed device and set-up for interrogating the sensor are detailed. Finally, results and conclusions are presented in Sections 5 and 6.

## 2. Sensor Configuration

Figure 1 shows a schematic of the proposed waveguide Bragg grating based MEMS quad-beam accelerometer with multi-axis sensitivity. The sensor consists of an amorphous-silicon (a-Si) waveguide with three Bragg gratings, Grating-1, Grating-2 and Grating-3 of different Bragg wavelengths such that their filter characteristics do not overlap, as shown in Figure 2. Grating-1 and Grating-3 are each integrated with two perpendicular suspending beam of the accelerometer.

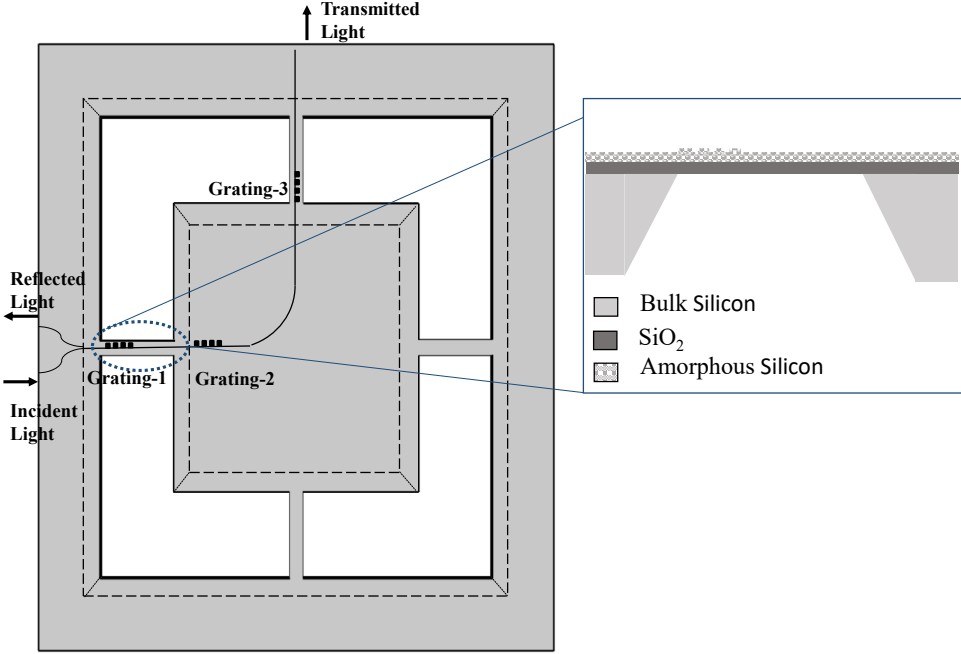

**Figure 1.** Schematic design of waveguide Bragg grating based Quad-Beam accelerometer: (Inset shows a cross-section view of one beam with amorphous-silicon(a-Si) waveguide and grating).

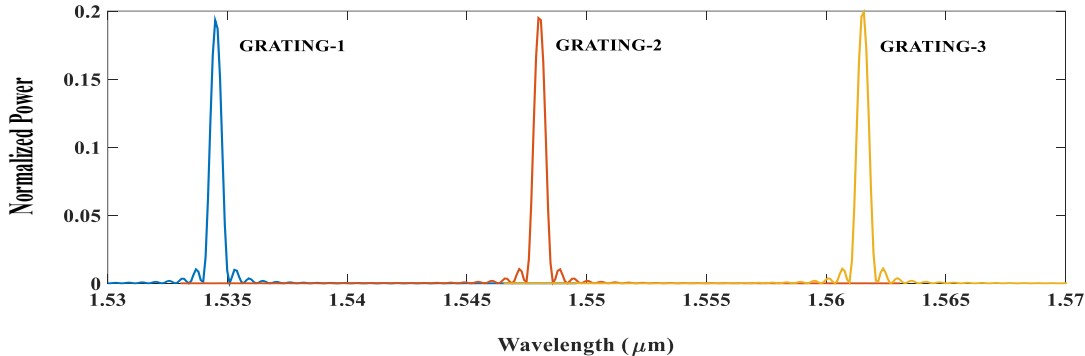

**Figure 2.** Reflection spectra of proposed sensor showing Bragg peaks of Grating-1, Grating-2 and Grating-3.

Grating-2 is integrated on the proof-mass and acts a reference for acceleration measurement. The Bragg gratings are in series configuration with reflected light showing three peaks as in Figure 2 when broadband input is launched. Light is launched into the device through an integrated directional coupler which acts as a circulator [42] to enable collection of reflected light as the output.

When the device frame is subject to a force, the proof mass displaces thus inducing stress in the suspending beams, which eventually results in a shift of Bragg wavelengths and reflection spectrum characteristics for Grating-1 and Grating-3. Depending on the direction of acceleration, the beams are either subject to compressional or tensile stress corresponding to blue or red shifts in the Bragg peaks. Grating-2 is insensitive to any external acceleration and acts as reference for noise cancellation due to variation of temperature experienced by the device.

## 3. Sensor Design and Mathematical Modelling

Figure 3a shows the side view of the portion of the sensor showing beam dimensions and compositions. The waveguide core material is taken as amorphous silicon (a-Si) based on the proposed fabrication procedure detailed in Section 4. A silicon wafer is used as substrate for the sensor instead of SOI platforms. The buried-oxide (BOX) layer thickness in typical SOI platforms is limited to below 5 μm. Also, the device layer silicon of such waveguides is thick (in range of 5–10 μm) and would require fine lapping, patterning and etching to realize the required dimensions. Instead, using a thermally oxidized silicon wafer of required thickness, a simple LPCVD process is sufficient to deposit silicon of the required thickness, which can then be easily patterned using IC photolithography. Hence, to simplify the fabrication process, the core material has been chosen as a-Si. Figure 3b shows typical stress and consequently the strain profile along beam length when subject to external acceleration. The sensor design and modelling will be discussed in following four subsections; mechanical properties of structure under acceleration studied using continuum mechanics, optical properties of waveguide Bragg grating due to stress and strain using coupled-mode theory, opto-mechanical coupling and sensitivity, cross-axis sensitivity and noise cancellation.

### 3.1. Mechanical Design of Quad-Beam Accelerometer

Figure 3a shows the beams suspending the proof mass to be composed of silicondioxide ($SiO_2$). Silicon proof mass is taken as a truncated pyramid with a square base, with a top side ($L_p$) 4 mm and thickness ($t_p$) 400 μm. A truncated pyramid is considered to account for bulk-micromachined silicon proof mass. The density of silicon is taken as 2329 kg/m³. $SiO_2$ has much lower Young's modulus and Poisson ratio than crystalline silicon. Strain induced in a material in response to applied stress/force is proportional to its Young's modulus. Poisson ratio gives information of a material's strain in a direction perpendicular to applied stress. Thus, a lower Young's modulus and Poisson ratio is preferred for a multi-axis accelerometer, as it would yield higher response and lower cross-axis sensitivity.

Under its own weight, the proof-mass of the quad-beam accelerometer displaces downwards inducing normal stress on the beam surface, $T_{\text{ow-normal}}(z)$ given by [11],

$$T_{\text{ow-normal}}(z) = \frac{3(l - 2z)}{2bh^2}mg \tag{1}$$

where $b, h, l$ are the width, thickness and length of the beam respectively, $g$ is the acceleration due to gravity and $m$ is the mass of the proof-mass. The length of the beam is considered to be along the z-direction with $z = 0$ representing the position near the foot of the beam. For lateral acceleration of $1g$ applied parallel to one set of beams and perpendicular to the other, the in-plane stress components for the same amount of force are $T_{\text{ow-}\parallel}(z)$ and $T_{\text{ow-}\perp}(z)$, which can be taken as [11],

$$T_{\text{ow-}\parallel}(z) = \frac{(t_p - h)}{bh^2 L_p}\left[\frac{(1.5L_{\text{p}} + l)}{(L_{\text{p}} + l)}l - 3z\right]mg \tag{2}$$

$$T_{\text{ow-}\perp}(z) = \frac{3(l - 2z)}{2hb^2}mg \qquad (3)$$

From the above equations, it is evident that beam geometry with a large width to height ratio is needed to minimize cross-axis sensitivity. The dimensions of proof-mass determine its mass and for maximizing the strain along beams, the mass must be maximized. Hence, based on practical considerations [47], the dimensions have been decided. The beams have a width of 210 μm and height 5 μm. The variation of all stress components along the length of the beam will follow the pattern of Figure 3b. As seen in Figure 3b, along the length of the beam suspending proof-mass, there is a change in sign of the strain (compressive to tensile or vice versa) with a portion of zero strain in the central region of the beam. The position where zero strain occurs and the amplitude and symmetry of stress distribution depends on the net magnitude and direction of acceleration experienced by the proof-mass.

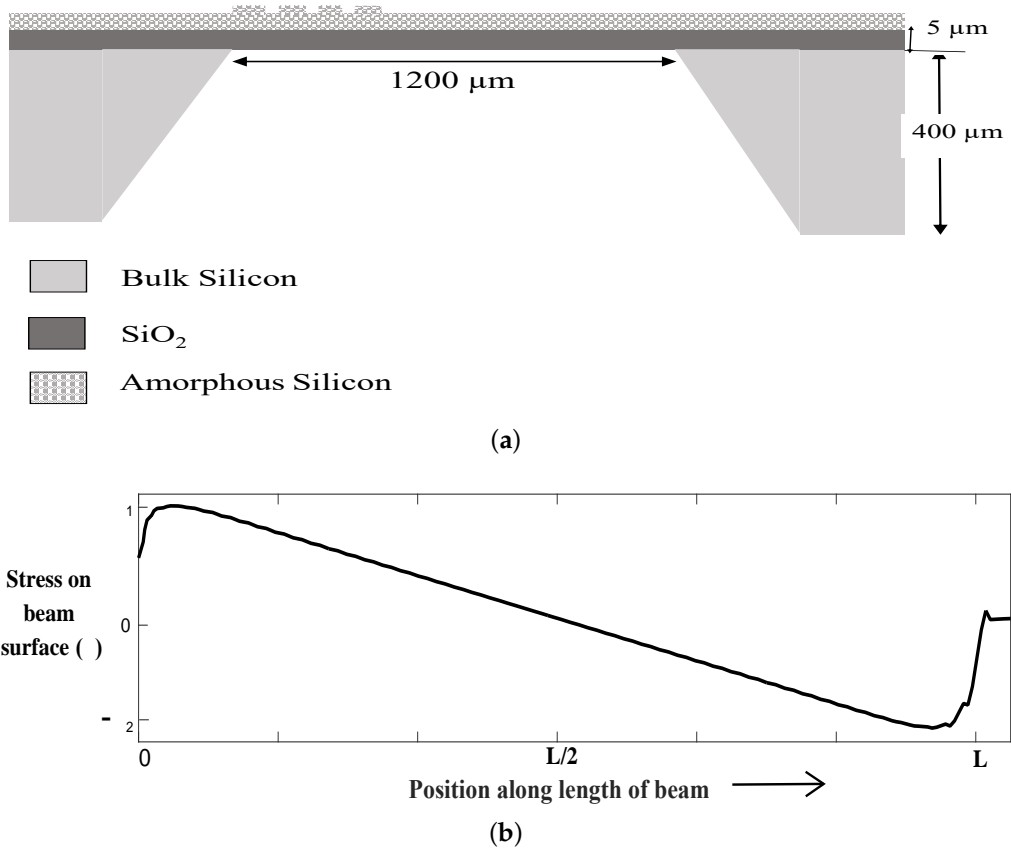

**Figure 3.** (**a**) Side view of portion of the sensor showing beam dimensions; (**b**) variation of stress along the length of the beam.

The actuation of the waveguide Bragg grating depends on the strain along its length. To realize a lateral accelerometer, an electrostatic repulsive force equal to the weight of proof-mass is applied to minimize the normal stress. This can be achieved by metallizing the bottom surface of the proof mass, bonding the device with another wafer carrying an electrode and maintaining the same potential on both surfaces using external electrical supply. The separation(*s*) between electrodes determines the capacitance between two surfaces (*C*) and thus the potential to be applied for compensating the effect of weight. The electrostatic force, $F_E$ for applied potential, $V$ is given by,

$$F_E = \frac{1}{2}V^2\frac{dC}{ds} \qquad (4)$$

Amorphous silicon on top of the SiO$_2$ layer represents the optical waveguide and has cross-section dimensions in only the order of ~4 μm$^2$ in contrast to the ~1000 μm$^2$ cross-section area of the SiO$_2$. By rule of mixtures for the composite beam of individual Young's modulus $E_1$ and $E_2$ having volume fractions $V_1$ and $V_2$ respectively, for stress applied laterally to the stack of materials, the effective moduli E$_{\text{eff}}$ is [48],

$$E_{\text{eff}} = \frac{E_1 E_2}{E_1 V_2 + E_2 V_1} \tag{5}$$

Since the volume contribution of silicon waveguide in the structure is very low, it has a negligible impact on the effective Young's modulus and thus the mechanical response of the device under acceleration.

The fundamental vibration frequency ($f$) of the sensor is for out-of-plane vibration and is found using Rayleigh's quotient as,

$$f = \frac{1}{2\pi}\sqrt{\frac{2E_{\text{eff}}bh^3}{ml^3}} \tag{6}$$

For the design values, $f$ is found as 92.43 Hz and verified by finite element analysis using COMSOL$^{\circledR}$ Multiphysics.

### 3.2. Waveguide Bragg Grating

Waveguide Bragg grating with surface-relief is designed for wavelengths around 1550 nm using TM mode for propagation. A rib waveguide with a core of a-Si height ($d$) of 1.5 μm (slab thickness is 0.8 μm and width 5 μm), waveguide width 1.6 μm runs over the SiO$_2$ beam of 5 μm thickness, which also acts as the cladding for the waveguide. The single-mode waveguide with the above dimensions satisfy the Soref condition [49] and have been reported earlier for realizing narrow bandwidth tunable Bragg grating based filters [50,51]. The slab of width 5 μm is considered so that it does not adversely impact either the optical or mechanical properties of the sensor. The refractive indices of the core and cladding are 3.545 and 1.45, respectively. The mode profile for fundamental mode propagating in this waveguide obtained by beam propagation method (BPM) is given in Figure 4. The dotted line in Figure 4 shows the transverse cross-sectional dimensions of the waveguide.

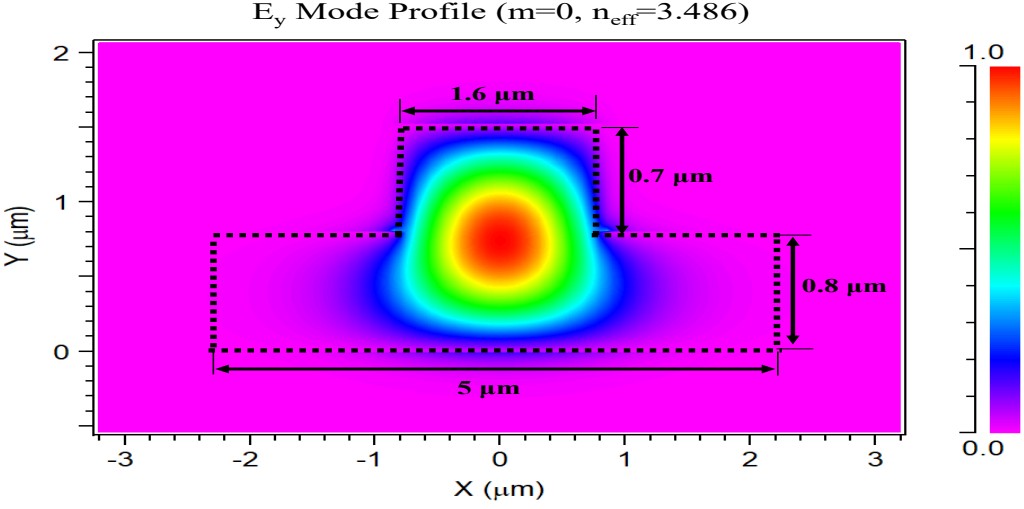

**Figure 4.** Mode profile of the waveguide (dotted line shows the cross-sectional dimensions of the waveguide core).

The longitudinal cross-section view of the Bragg grating is shown in Figure 5. For a grating of period Λ, the Bragg wavelength, $\lambda_B$ is (7)

$$\lambda_B = 2n_{eff}\Lambda \tag{7}$$

where $n_{eff}$ is the effective refractive index of the waveguide calculated using waveguide analysis [52].

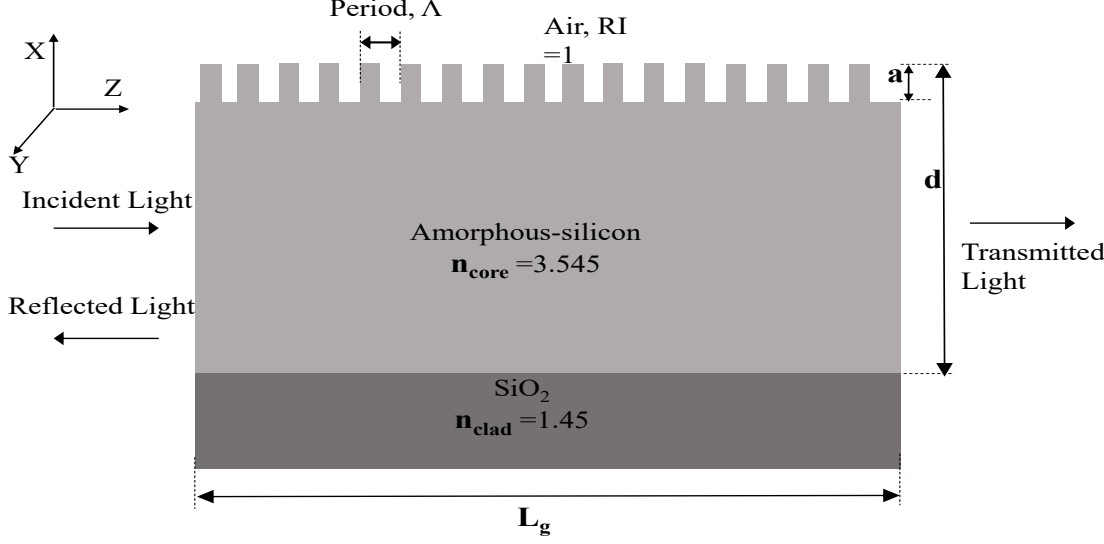

**Figure 5.** Longitudinal cross-section view of surface-relief waveguide Bragg grating.

As light propagates along the grating, due to the periodic corrugations, partial reflection occurs at each period. For light in which the wavelength satisfies the Bragg condition, the reflected and transmitted fields are in phase and thus the backward and forward propagating waves couple. The theoretical determination of this is performed using coupled mode theory [53,54]. Z-direction is the direction of propagation of light with grating boundaries at $z = 0$ and $z = L_g$. For all calculations, only the first order diffraction and fundamental mode have been considered. Coupled mode equations of a Bragg grating with $R$ and $S$ as complex-amplitudes of forward-running and backward-running modes considered for analysis are (8) and (9).

$$\frac{dR}{dz} = i\hat{\sigma}R(z) + i\kappa S(z) \tag{8}$$

$$\frac{dS}{dz} = -i\hat{\sigma}R(z) - i\kappa^* S(z) \tag{9}$$

where the coupling coefficient, $\kappa$ and $\hat{\sigma}$ are given by

$$\hat{\sigma} = \frac{\pi}{\lambda}\delta n_{\text{eff}} + 2\pi n_{\text{eff}}\left(\frac{1}{\lambda} - \frac{1}{\lambda_B}\right) - \frac{1}{2}\frac{d\phi}{dz} \tag{10}$$

$$\kappa = \frac{2\pi^2 a^3}{3\lambda_B d^3}\left(\frac{n_{\text{core}}^2 - n_{\text{clad}}^2}{n_{\text{core}}^2}\right)k_{c^*} \tag{11}$$

$$k_{c^*} = \left[1 + \frac{3\lambda_B}{2a\pi\sqrt{(n_{\text{core}}^2 - n_{\text{clad}}^2)}} + \frac{3\lambda_B^2}{4a^2\pi^2(n_{\text{core}}^2 - n_{\text{clad}}^2)}\right] \tag{12}$$

where $\frac{1}{2}\frac{d\phi}{dz}$ is a z-dependent phase term that describes the chirp of the grating period. For uniform period grating without such a variation, the reflected power spectrum $r(\lambda)$ is obtained as

$$r(\lambda) = \frac{sinh^2(\sqrt{\kappa^2 - \hat{\sigma}^2}L_g)}{cosh^2(\sqrt{\kappa^2 - \hat{\sigma}^2}L_g) - \frac{\hat{\sigma}^2}{\kappa^2}} \tag{13}$$

For a uniform grating, the full-width-half-maximum (FWHM) bandwidth $\Delta\lambda_{FWHM}$ is given by [54]

$$\Delta\lambda_{FWHM} = s\lambda_B\sqrt{\left(\frac{\delta n_{eff}}{2n_{eff}}\right)^2 + \left(\frac{\Lambda}{L_g}\right)^2} \tag{14}$$

where $s$ is taken as 1 for strong gratings ($r \approx 1$) and 0.5 for weak gratings since the band edge of the grating spectrum has very sharp roll-off for strong gratings. From (14) it is clear that for strong gratings characterized by high modulation depth in grating, the dominant term is the first term of the square-root. Hence, FWHM of strong gratings is independent of length and depends only on modulation depth. In contrast, for weak gratings, the second term of (14) dominates and FWHM decreases with increase in grating length ($L_g$). Performance of Bragg grating as a sensor depends on accurate determination of the peak of the spectrum [55]. Lowering the modulation depth would decrease the spectral bandwidth and improve sensitivity and resolution of the sensor [56]. However, fabrication of grating profile with accurate and low etch depth is very challenging. Grating modulation depth ($a$) of 50 nm has been considered in the design based on values reported in the literature [42]. Further, in long weak gratings the reflected power is low thus requiring a sensitive intensity detection set-up. Thus, as a trade-off, for the proposed device, $L_g$= 600 µm is chosen based on the optical and mechanical constraints.

However, in the presence of chirp or z-dependence of the period, the solutions to the coupled mode equations are evaluated numerically by formulating it as Riccati differential equation. When there is linear variation in period over grating length or linear chirp, there is a spectral broadening of the grating characteristics, which can be specified in terms of chirp parameter $F$[57] where,

$$\phi(z) = \frac{Fz^2}{L_g{}^2} \tag{15}$$

and $F$ parameter is

$$F = 4\pi n_{eff}\frac{d\lambda_B}{dz}\frac{\Delta\lambda_{FWHM}}{\lambda_B{}^2} \tag{16}$$

where $\frac{d\lambda_B}{dz}$ gives variation of Bragg wavelength between ends of the grating. The numerical solution of the reflected spectrum of the chirped grating shows that the width of the main lobe increases to overlap with side lobes in which the peak power also increases. The results obtained from coupled mode solutions using RSoft® Component Design Suite's GratingMOD tool match the expected outcomes for linearly chirped gratings.

### 3.3. Opto-Mechanical Coupling

The Bragg wavelength shift in a grating is due to both a change in the local period of the grating and change in the effective refractive index, which can be mathematically written as

$$\Delta\lambda_B = 2(\Lambda\Delta n_{eff} + n_{eff}\Delta\Lambda) \tag{17}$$

When subject to external lateral acceleration, longitudinal and lateral stresses are induced in the beams. The local period of the grating positioned on a beam depends on the longitudinal strain on the beam surface. As a-Si forms the core of the waveguide, the photoelastic effect of variation in effective index is negligible. The photoelastic effect generally plays a significant role in the design of optical

MEMS [58]. The residual stresses induced in materials of different layers in MEMS structures resulting from different fabrication processes affect the optical properties (refractive index, birefringence) due to change in structural symmetry at the crystal level. Further, variations of temperature in working conditions also cause a photoelastic effect. However, the photoelastic effect on the sensor will be negligible because of the dimensions of the structures involved. The impact of the photoelastic effect on the optical properties (both linear and $\chi^2$ processes) is found to reduce with film thickness and for values greater than 2.5 μm it can be neglected [59]. However, a change in effective index is possible due to the thermo-optic coefficient ($\frac{dn}{dT}$). From Figure 3b, it is clear that the strain varies linearly along the grating length, thus making the grating linearly chirped. However, a longer grating on the beam would also imply that some portions of the grating experiencing positive chirp and the other negative chirp. For maximizing the sensitivity of the device, the gratings must see maximum strain. Thus, the gratings need to be positioned appropriately along the beam length. As shown in Figure 1, Grating-1 is from the frame end to the centre of the beam and Grating-3 is positioned from the proof-mass end to the centre of a perpendicular beam. Grating-2 positioned on the proof mass is not affected by external strain.

*3.4. Sensitivity and Resolution*

The difference between reflection spectrum peaks of Grating-1 and Grating-2, is the measure of the horizontal component of in-plane acceleration, and the difference between Grating-1 and Grating-3 is the measure of the vertical component of in-plane acceleration. From Equations (2) and (3) for the proposed device the ratio of maximum stresses induced in parallel and perpendicular beams for a unit force applied parallel to a beam is,

$$\frac{(T_\parallel)_{\text{max}}}{(T_\perp)_{\text{max}}} = \frac{2b(t_p - h)(1.5L_p + l)}{3L_p h(L_p + l)} = 3.8285 \tag{18}$$

With the Poisson ratio of SiO$_2$ being 0.17, the maximum longitudinal strain due to acceleration in a direction perpendicular to the beam is 4.44% of the strain for the same acceleration parallel to the length of the beam. Further, due to the distributed sensing nature of the Bragg grating, the strain values experienced by the grating are even lower. Thus, the cross-axis sensitivity of the proposed sensor is even lower.

The sensitivity of the sensor depends on the accurate determination of the peak of the reflected spectrum. Prominent source of noise impacting the grating output is temperature variation. The temperature sensitivity, K$_T$ of the Bragg gratings depends on Bragg wavelength, thermal expansion co-efficient- $\alpha$ and thermo-optic co-efficient- $\xi$ as

$$K_T = \lambda_B(\alpha + \xi) \tag{19}$$

$\xi$ depends on the ambient temperature and the operating wavelength. It is about $1.5 \times 10^{-4}$/K when the device is operated at around room temperature [60]. $\alpha$ for silicon is $2.62 \times 10^{-6}$/K. Thus, temperature sensitivity of gratings is estimated as 236.56 pm/K. If K$_{a1}$, K$_{a2}$ and K$_{a3}$ are acceleration sensitivities of Grating-1, Grating-2 and Grating-3, respectively, the Bragg wavelength shifts can be written as

$$\Delta\lambda_{B1} = \Delta T K_T + \Delta a K_{a1} \tag{20}$$
$$\Delta\lambda_{B2} = \Delta T K_T \tag{21}$$
$$\Delta\lambda_{B3} = \Delta T K_T + \Delta a K_{a3} \tag{22}$$

Thus, the components of lateral acceleration along directions of Grating-1 and Grating-3 can be calculated by taking the difference of the respective Bragg wavelength shifts from the Bragg shift of Grating-2.

In the proposed sensor, since the light remains guided and the acceleration measurement is done based on the peak position and not the intensity, the dominant noise mechanism limiting the resolution of the device is mechanical-thermal noise. The noise-equivalent acceleration $a_{th}$ can be estimated as [61]

$$a_{th} = \sqrt{\frac{8\pi K_B T f_o}{mQ}} \tag{23}$$

where $K_B$ is the Boltzmann's constant (in N-m/K), $T$ is the temperature of the sensor (in K), $f_o$ is the fundamental resonance frequency (in Hz), $m$ is the mass of proof-mass (in Kg) and $Q$ is the mechanical Q-factor. The mechanical Q-factor is determined by the sharpness of resonance in the frequency response of the device [62]. The sharpness of the resonance depends on the damping factor ($\zeta$) of the system. The Q-factor is given as $Q = 0.5/\zeta$. As under-damped conditions result in a ringing response, most accelerometers are operated at critical damping ($\zeta = 1$) to maximize bandwidth with good sensitivity. The noise-equivalent acceleration, $a_{th}$ is the measure of resolution of the sensor.

## 4. Proposed Fabrication Procedure and Testing Set-up

Figure 6 shows a possible fabrication method for the proposed device. After initial wafer cleaning, thermal oxidation is done to realize SiO$_2$ of required thickness as in step-B. Low pressure chemical vapour deposition (LPCVD), as shown in step-C, is suitable for deposition of amorphous silicon (polysilicon), which forms the core of the waveguide and Bragg grating. The amorphous-silicon will form the waveguide core with the SiO$_2$ forming the under-cladding. Photolithography and reactive-ion-etch (RIE) of the amorphous silicon layer is done to define the waveguide, input and output ports of directional coupler and Bragg gratings as shown in step-D. Use of RIE etch would ensure realizing device dimensions with small tolerances and also good sidewall smoothness [63]. Also, the top surface of the quad-beams are defined and patterned at this stage. Then, a sacrificial photo-resist (PR) layer is deposited to protect the waveguide before flipping the wafer as in steps-E,F. Then, the exposed bottom surface of the substrate silicon is metallized, patterned and subject to bulk wet etch using KOH or TMAH to release the proof mass. Finally, the deposited photo-resist (PR) is stripped to reveal the sensor. The device can then be completed by bonding with another chip with a metallized surface to form electrodes for electrostatic actuation.

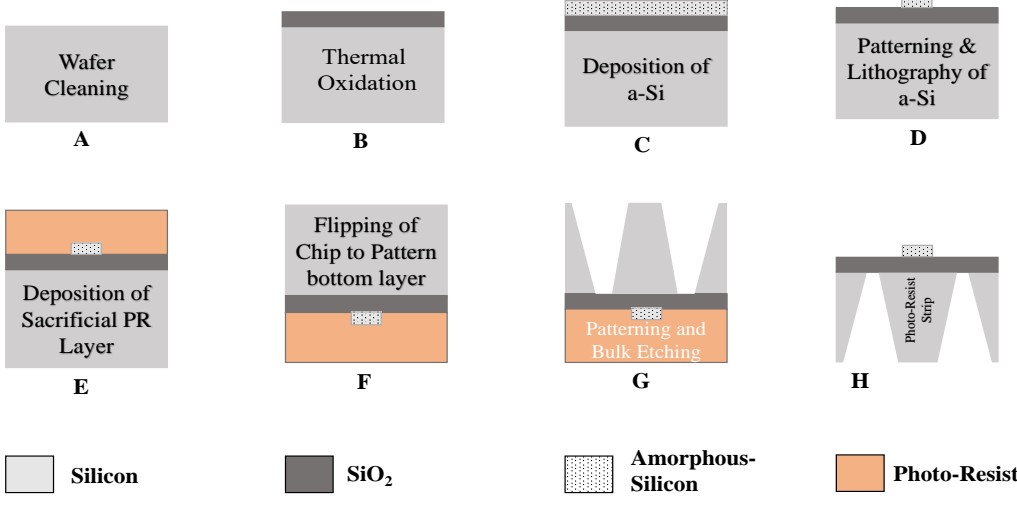

**Figure 6.** Proposed fabrication procedure.

There are many interrogation methods available for detecting the Bragg peak [33,64]. The interferometric scanning method based on path unbalanced Mach–Zehnder interferometer [65] provides resolution in the order of nano strain ($10^{-3}$ $\mu\epsilon$) and would be appropriate for detection as the resolution of the proposed sensor can reach ~1 ng/$\sqrt{\text{Hz}}$. This interrogation scheme does not require

ultrastable or tunable lasers. Further, a single path unbalanced Mach–Zehnder interferometer has been demonstrated to act as a multispectral wavelength shift detector [66]. Since planar waveguides exhibit polarization dependent properties, a superluminescent diode (SLD) at 1550 nm (for example SLD-761-HP from SUPERLUM) is taken through a polarization controller (PC) and used as the optical source as shown in Figure 7. The proposed testing set-up involves wavelength demultiplexing of the reflected grating spectra of each grating and extraction of phase shift information using phase generated carrier (PGC) modulation of phase in the Mach–Zehnder interferometer. The acceleration is then deduced from the difference of peak shifts between the sensitive grating (Grating-1 and Grating-3) peaks to the reference peak (Grating-2).

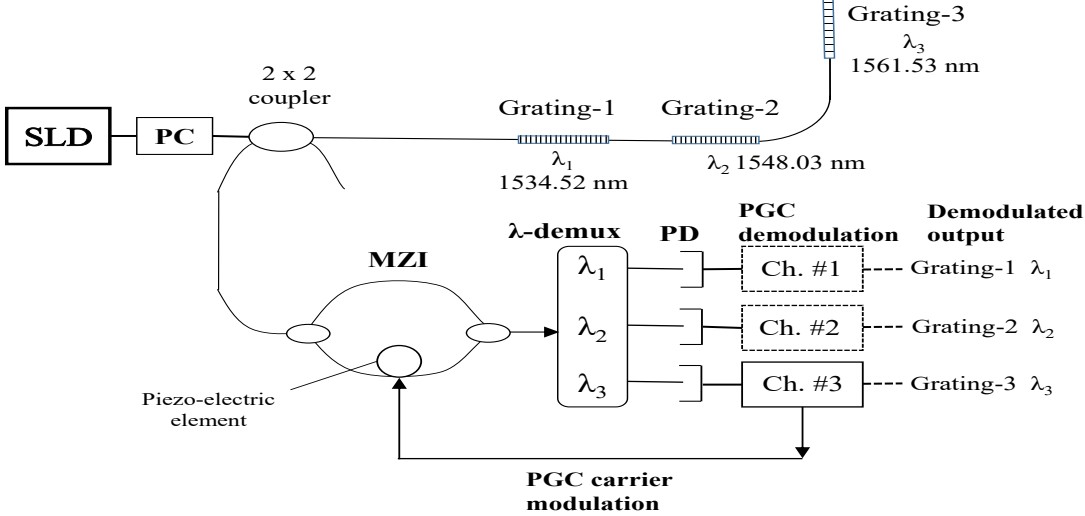

**Figure 7.** Proposed testing set-up using wavelength demultiplexer and unbalanced Mach–Zehnder interferometer.

## 5. Results and Discussion

The lateral quad-beam waveguide Bragg grating accelerometer has been simulated using COMSOL$^{\circledR}$ Multiphysics and RSoft$^{\circledR}$ GratingMOD. The dimensions of the device and material properties used for calculations are summarized in Table 2.

**Table 2.** Device dimensions and material properties.

| Parameter | Value |
|---|---|
| Top side length of square proof-mass ($L_p$) | 4 mm |
| Thickness of proof-mass ($t_p$) | 400 μm |
| Length of each beam ($l$) | 1200 μm |
| Width of each beam ($b$) | 210 μm |
| Thickness of each beam ($h$) | 5 μm |
| Length of each Bragg grating ($L_g$) | 600 μm |
| Etch depth of Bragg grating ($a$) | 50 nm |
| Refractive index of core in operating wavelength range ($n_{core}$) | 3.545 |
| Refractive index of cladding in operating wavelength range ($n_{clad}$) | 1.45 |
| Thermo-optic coefficient at room temperature ($\xi$) | $1.5 \times 10^{-4}$/K |
| Thermal expansion coefficient ($\alpha$) | $2.62 \times 10^{-6}$/K |

In rest condition, Grating-1, Grating-2 and Grating-3 have periods 220, 222 and 224 nm, respectively. Corresponding Bragg wavelengths are 1534.52, 1548.03 and 1561.53 nm, respectively. The FWHM of each grating spectrum is less than 570 pm. Figure 8 shows the variation of strain along the length for acceleration parallel to the length of one of the beams of the quad-beam accelerometer for different accelerations. Similarly, Figure 9 shows variation of strain along the length for acceleration perpendicular to the beam length for different accelerations.

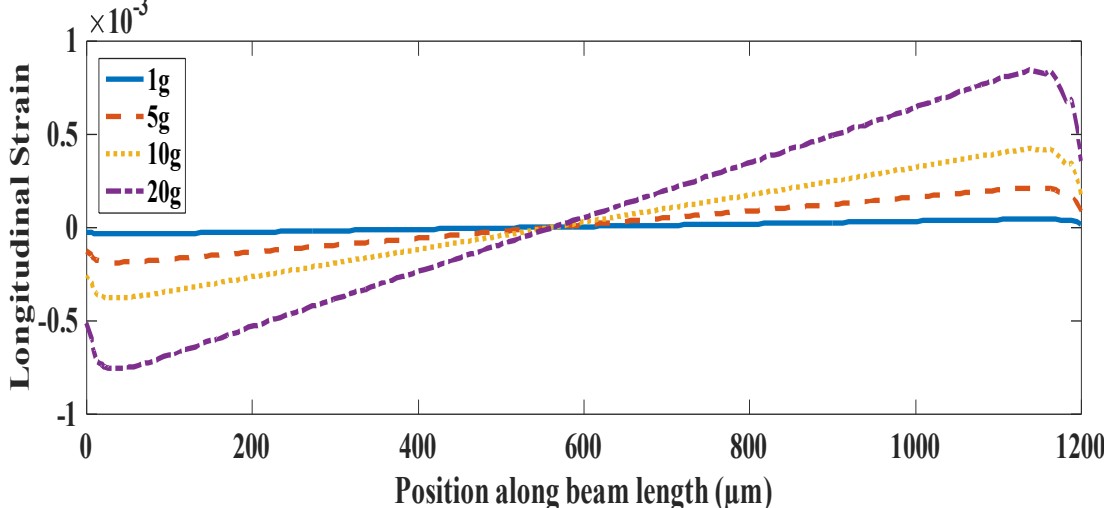

**Figure 8.** Longitudinal strain for different parallel accelerations.

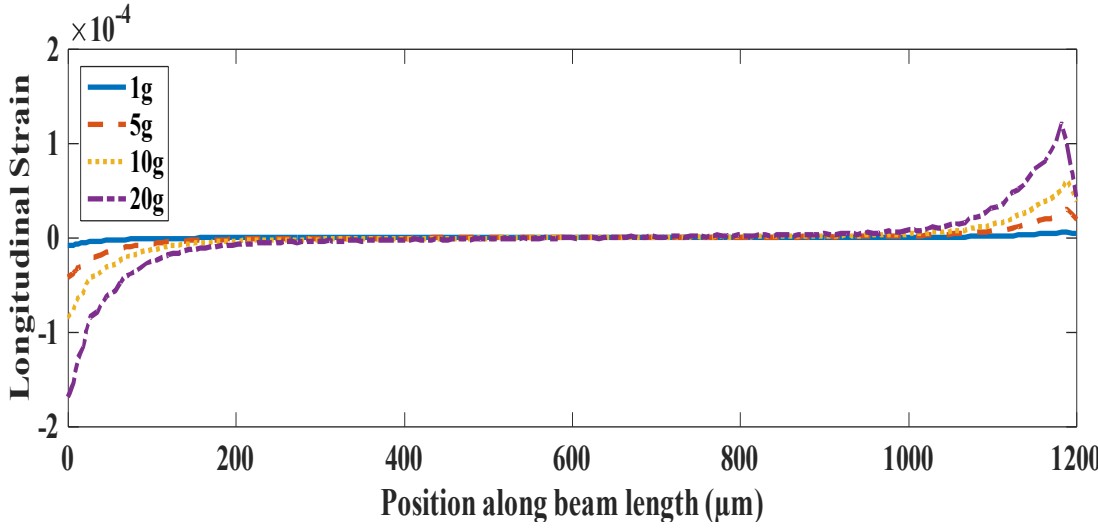

**Figure 9.** Longitudinal strain for different perpendicular accelerations.

The result in Figure 9 is evidence that the effect of lateral acceleration in the direction perpendicular to the beam length has very little impact on the sensor output. The reflected spectra for different lateral accelerations parallel to the beam carrying Grating-3 are shown in Figure 10. Plotting spectral peaks against corresponding acceleration as in Figure 11 shows that the grating peak varies linearly with applied acceleration. Sensitivity of the grating is found to be 30 pm/$g$. Comparing this to the temperature sensitivity of 236.56 pm/K, we can observe that in absence of suitable temperature compensation there will be a large error in sensor output. Hence, in the proposed design Grating-2, a reference grating has been introduced and positioned on the proof mass such that its' spectral properties are unaffected by acceleration experienced by the device. Calculating differential peak

shifts with respect to reference grating Grating-2 will mitigate the undesirable effect of temperature variations on sensor performance.

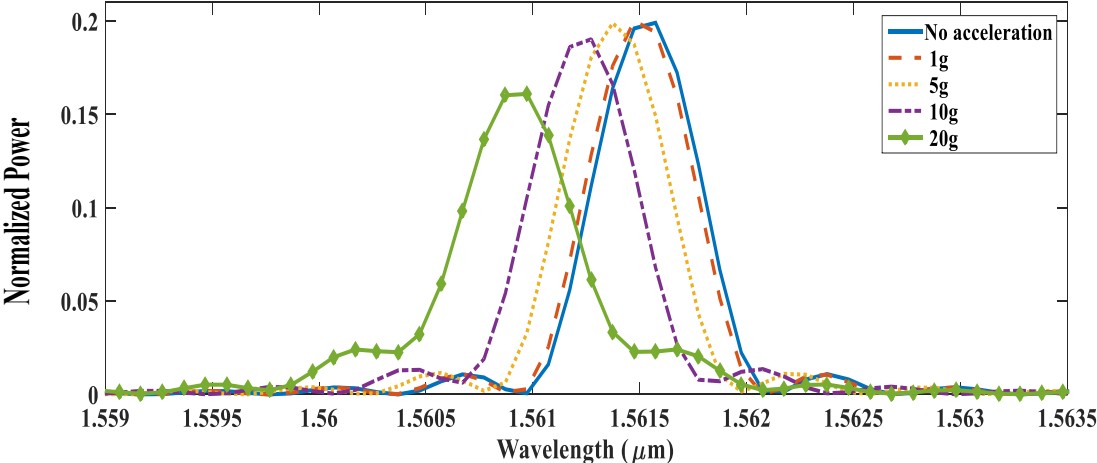

**Figure 10.** Reflection spectra for Grating-3 for different parallel accelerations.

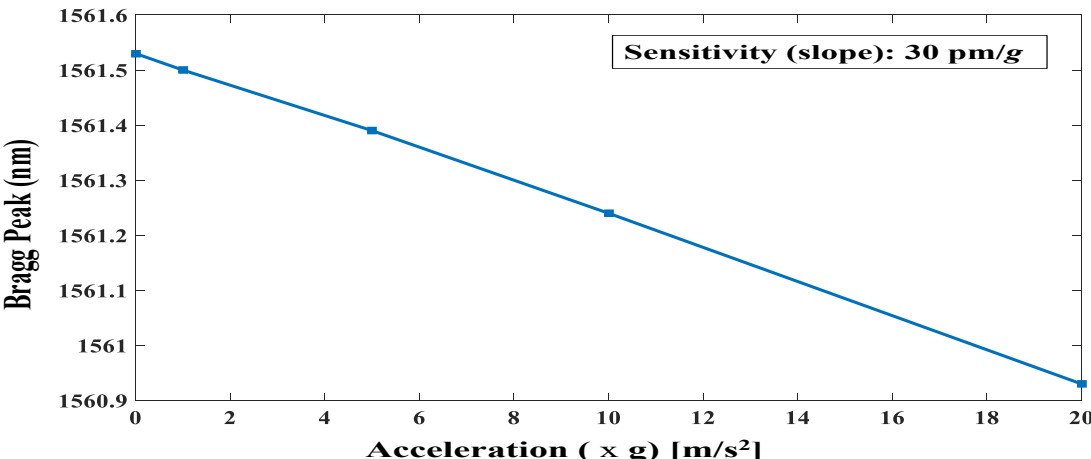

**Figure 11.** Variation of Bragg peak with applied parallel acceleration causing compression of Grating-3.

As the acceleration is causing compression in the grating, with increasing acceleration there is a blue shift of the peak of the reflected spectrum. Hence, any in-plane acceleration can be accurately measured in terms of its Cartesian coordinate components in the plane oriented along the beams. From Figure 10, it is observed that with increasing actuation, there is a spectral flattening (FWHM for 20 g increases to 600 pm) alongside reduction in peak power. This flattening ultimately limits the range of the sensor.

Fundamental frequency of the proposed sensor is 92.43 Hz due to large dimensions of the proof-mass. The resolution of the sensor, determined by the noise-equivalent acceleration is obtained using Equation (23) for critical damping (Q = 0.5) is 0.116 $\mu g/\sqrt{Hz}$ at room temperature. Even when the temperature is raised by one hundred Kelvin, the resolution becomes 0.134 $\mu g/\sqrt{Hz}$. In various high-resolution accelerometer experiments [19,22] mechanical Q-factor in the order of $10^5$ or higher has been achieved in vacuum. The accelerometers in [19,22] with only one sensing axis each have resolutions in the range of ~10 $\mu g/\sqrt{Hz}$. In comparison, for a Q-factor of $10^5$, resolution of the proposed design is a remarkable 0.26 $ng/\sqrt{Hz}$. The high-resolution and large range of the proposed lateral 2-axis accelerometer clearly show its suitability for inertial navigation and seismometry.

## 6. Conclusions

A novel high-resolution optical MEMS quad-beam lateral dual-axis accelerometer based on waveguide Bragg gratings has been proposed and analyzed in this work. Coupled mode theory is used to design and analyze the a-Si/SiO$_2$ based waveguide Bragg gratings, which are the sensing elements. Quad-beam structure and materials have been optimally chosen to maximize the longitudinal strain and thus the response of the sensor. A reference grating Grating-2 has been used to eliminate the effect of temperature variation on the device, which otherwise could have had a grave impact on sensor performance. The difference in peaks of the reflected spectrum of Grating-1 and Grating-3 with respect to Grating-2 provide measure of in-plane acceleration in terms of its Cartesian components. To facilitate higher sensitivity, out-of-plane acceleration resulting from weight of proof-mass is compensated using electrostatic force. A linear sensitivity of 30 pm/g with large range and cross-axis sensitivity lower than 1% is obtained with the proposed sensor. A high-resolution of less than 1 μg/$\sqrt{\text{Hz}}$ can be obtained with the proposed sensor, thus making the proposed design suitable for inertial navigation and seismometry applications.

**Author Contributions:** Conceptualization, B.M.; formal analysis, B.M. and P.K.P.; writing—original draft, B.M.; writing—review and editing, N.K., and P.K.P. All authors have read and agreed to the published version of the manuscript.

**Funding:** This research received no external funding

**Conflicts of Interest:** The authors declare no conflict of interest.

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
