# Peer review of "Novel High-Resolution Lateral Dual-Axis Quad-Beam Optical MEMS Accelerometer Using Waveguide Bragg Gratings"

_photonics, doi:10.3390/photonics7030049_

Round 1

Reviewer 1 Report

This manuscript reports on the design and modelling of quad beam accelerometer with amorphous Si on SiO2 Bragg gratings to enable integrated optical readout with high sensitivity and resolution. It is shown that with optimum design parameters the device provides a linear response over a large range with high resolution. I recommend the manuscript for publication after addressing these comments:

Q1) In the introduction, it discusses other works that have proposed or demonstrated MEMS devices with PIC components such as photonic crystal cavities, microdisk, etc. Can you please provide a more detailed comparison of the benefits/disadvantage of your approach of using Bragg grating. A table would be very useful for readers of the manuscript.

Q2) From Fig. 2 it appears you need an optical source that is fairly broadband. What is the intended light source for this device to operate? Depending upon the design Bragg gratings are normally polarization sensitive so an LED might not be suitable.

Q3) In Fig. 3 (b) is difficult to interpret with no y-axis values. Is this plot showing that the beam is subject to both compressive and tensile strain along the beam length?

Q4) Why is amorphous Si used as the waveguide core and not a more typical silicon photonic platform such as SOI?

Q5) Is amorphous Si etched by standard Si micro-machining processes such as KOH and TMAH? If so, could the authors please comment on how this device should be fabricated.

Q6) Why was a 1.3 um x 0.77 um core waveguide dimension chosen? Will this not be significantly multi-modal, does that pose any challenge to the device operation?

Q7) What is the etch depth of the grating? Could you not make the grating more weakly coupled (smaller etch depth) to enable longer grating lengths and still have a narrow bandwidth response and sufficient reflection?

Q8) How do you measure the Bragg wavelength shift with acceleration?

Q9) From Fig. 9 it shows the Bragg response being less than a nanometre for 20 g acceleration. I’m guessing you might need a narrow-linewidth tunable laser rather than a grating spectrometer?

Reviewer 2 Report

The article “Novel High-Resolution Lateral Dual-Axis Quad-beam Optical MEMS Accelerometer using Waveguide Bragg Gratings” by Balasubramanian Malayappan, Narayan Krishnaswamy and Prasant Kumar Pattnaik is a theoretical work regarding an accelerometer integrated with bragg gratings for the readout.

This paper is well written and didactically presented. However, the reference list is not present. For reviewers it is very difficult to evaluate the introduction without this list, except for authors previous papers. Please fix this issue. Maybe additional references are needed.

The paper is inserted in the authors research topics. The sensor described is based on a very common architecture. The novelty elements are the Bragg gratings. Bragg sensors integrated in mems and fibers are very sensitive devices (see for example Gagliardi et al papers). In this context, similar (or even better) performances could be achieved by gratings integrated in fiber sensors using ultrastable laser. However the costs increase.

The strong point of this paper is the carefully modeling of the structure.

I have only few comments in order to improve the overall quality of the manuscript.

-I suggest to include a discussion about fabrication methods and in particular in which way these processes could afflict the model (i.e. influence or wet etching or dry etching on pattern and consequences on geometry and on parameters, a, d, Λ)

-In figure 2 authors report three Bragg peaks for their sensors. I suggest to discuss about the FWHM of these peaks. It would be possible to obtain narrow peaks? This could be increase sensitivity of the sensors.

-In addition, I suggest to the authors to add a discussion or a detailed picture of the readout scheme. It is not really clear if all the peaks would be acquired in reflection. In which way the authors would like to discriminate the three peaks? Scanning the laser?

-What do the authors mean with the directional coupler on line 80? A fiber circulator?

-There are some typos to fix

Reviewer 3 Report

The authors present their design of dual-Axis optical MEMS accelerometer. Main aspects of optical MEMS accelerometer have been considered in the design. The paper is very well organized and presented. Unfortunately there is no any experiment in the work to justify the design. The received manuscript seems incomplete: there is no bibliography included, which might hamper the understanding of some points in the work.

The following comments are for authors' considerations:

  1. In line 156: “Increasing length of the grating, Lg increases the reflected power but also causes broadening of spectrum”. This statement is not entirely true. For small coupling coefficients, the grating bandwidth decreases with increasing grating length while peak reflection increases, until peak reflection saturates. Similarly, a grating with small length could reflect enough power with large coupling coefficients, at the cost of broadening spectrum.
  2. In line 176: “As a-Si forms the core of the waveguide, the photoelastic effect of variation in effective index is negligible.” The authors should give more comments on this point as the photoelastic effect in optical MEMS always is a significant issue. The large difference between core, cladding and wafer consequently leads to the variation of photoelastic effect with temperature which causes more problems in real applications.
  3. In the design, “Grating-1 is from the frame end to the centre of the beam and Grating-3 is positioned from the proof-mass end to the centre of a perpendicular beam.” It is easy to understand that grating-1 is positioned to achieve maximum sensitivity. In the case grating-2 should be positioned in a similar way because it is 2-axis sensor and is supposed to measure the acceleration not limited in the direction parallel to grating-1. However, grating-2 is positioned to receive the minimum sensitivity in its own direction, which doesn’t mean a minimum cross sensitivity. For minimum cross sensitivity, according to Fig. 7, the grating should be positioned at the centre of the beam (unless I misunderstood the figure).
  4. Since there is no bibliography the reviewer could not know how the accelerometer resolutions in [16,17] are achieved. However, I would not compare the resolution of a theoretical design with that of experimental sensor as they are different in many aspects.

Round 2

Reviewer 1 Report

The authors have kindly addressed all my previous comments and concerns, therefore, I am happy to recommend the manuscript for publication. 

Reviewer 2 Report

The manuscript has been revised. The readability of the manuscript has been enhanced.

The introduction has been improved and the references list is adequate.